# Cognitive map formation under uncertainty via local prediction learning

## Abstract

Cognitive maps are internal world models that enable adaptive behavior including spatial navigation and planning. The Cognitive Map Learner (CML) has been recently proposed as a model for cognitive map formation and planning. A CML learns high dimensional state and action representations using local prediction learning. While the CML offers a simple and elegant solution to cognitive map learning, it is limited by its simplicity, applying only to fully observable environments. To address this, we introduce the Partially Observable Cognitive Map Learner (POCML), extending the CML to handle partially observable environments.

The POCML employs a superposition of states for probabilistic representation and uses binding operations for state updates. Additionally, an associative memory is incorporated to enable adaptive behavior across environments with similar structures. We derive local update rules tailored to the POCML's probabilistic state representation and associative memory. We demonstrate a POCML is capable of learning the underlying structure of an environment via local next-observation prediction learning. In addition, we show that a POCML trained on an environment is capable of generalizing to environments with the same underlying structure but with novel observations, achieving good zero-shot next-observation prediction accuracy, significantly outperforming sequence models such as LSTMs and Transformers. Finally, we present a case study of navigation in a two-tunnel maze environment with aliased observations, showing that a POCML is capable of effectively using its probabilistic state representations for disambiguation of states and spatial navigation.

## 1 Introduction

Cognitive maps are central to the adaptive behavior of intelligent agents, enabling capabilities ranging from spatial navigation to planning to reasoning and abstraction; they are internal world models that allow agents to predict the consequences of their actions (Behrens et al., 2018). Known to be responsible for spatial representations in the brain (Eichenbaum, 2017; Behrens et al., 2018), the hippocampal formation (HF), consisting of the hippocampus and the entorhinal cortex, is a primary source of inspiration for models of cognitive map learning. While we do not know how exactly the brain implements cognitive maps, there are various theoretical models of how this might be done at an algorithmic level (Whittington et al., 2020; George et al., 2021; Stöckl et al., 2024).

Recently, Stöckl et al. (2024) proposed the Cognitive Map Learner (CML), a simple model that can learn high-dimensional representations of states and actions that reflect the structure of the underlying environment using local synaptic plasticity rules that minimize next-state prediction error. Using these learned representations, a CML can subsequently perform online planning to reach a goal state via a simple action selection heuristic; i.e. by choosing the action most similar to the vector formed by the difference between the goal state and current state. Compared to probabilistic models, such as the Tolman-Eichenbaum machine (TEM) (Whittington et al., 2020) and clone-structured cognitive graph (CSCG) (George et al., 2021), CML is more computationally efficient, geometrically interpretable, and only requires local learning rules to train.

However, CML is limited by its simplicity, applying only to fully observable environments where observations unambiguously distinguish different states. In comparison, models such as TEM and

CSCG are more expressive and can handle partially observable environments due to their probabilistic nature; that said, they are limited by their computational cost and limited interpretability.

To fill in this research gap, we introduce the Partially Observable CML (POCML), an extension of a CML to partially observation environments. A POCML leverages random Fourier features to enable the representation of a state in superposition. The superposition of states is used to represent states in a probabilistic manner, to which a binding operation is used for updating these states simultaneously and independently. To decouple the underlying environment structure and state-observation relation, we endow the POCML with a memory storing state-observation associations to enable adaptive behavior in different environments with the same underlying structure and derive corresponding local update rules that take the new probabilistic state representation and associative memory into consideration.

We demonstrate that when presented with sequences of actions and observations of random walks in an environment, a POCML is capable of learning the underlying structure via local next-observation prediction learning. In addition, we show that a POCML trained on an environment is capable of generalizing to environments with the same underlying structure but with novel observations, achieving good zero-shot next-observation prediction accuracy, significantly outperforming sequence models such as LSTMs (Hochreiter & Schmidhuber, 1997) and Transformers (Vaswani et al., 2017). Finally, we present a case study of navigation in a two-tunnel maze environment with aliased observations, showing that a POCML is capable of effectively using its probabilistic state representations for disambiguation of states and spatial navigation.

## 2 BACKGROUND

### COGNITIVE MAP LEARNERS

Given a sequence of observations $\mathbf{o}_t$ and actions $\mathbf{a}_t$ produced by an agent acting in an environment, a CML learns to predict the next observation based on the current observation and action taken by the agent (Stöckl et al., 2024).

In a CML, observations $\mathbf{o}_t \in \mathbb{R}^{n_o}$ and actions $\mathbf{a}_t \in \mathbb{R}^{n_a}$ are represented as one-hot vectors and are embedded into a common high dimensional state space $\mathcal{S} \subseteq \mathbb{R}^n$ via embedding matrices $\mathbf{Q} \in \mathbb{R}^{n \times n_o}$ and $\mathbf{V} \in \mathbb{R}^{n \times n_a}$, respectively. Here, $n_o$ is the number of observations, $n_a$ is the number of actions, and $n$ is the dimension of the state space. Both $\mathbf{Q}$ and $\mathbf{V}$ have entries sampled i.i.d. from $\mathcal{N}(0, 1)$ and are then normalized by $1/\sqrt{n}$.

Given an observation, the corresponding state representation is

$$\mathbf{s}_t = \mathbf{Q}\mathbf{o}_t \tag{1}$$

and, in addition, given an action, the predicted next state is

$$\hat{\mathbf{s}}_{t+1} = \mathbf{s}_t + \mathbf{V}\mathbf{a}_t. \tag{2}$$

We want the state and action representations to be such that such that predicted next state $\hat{\mathbf{s}}_{t+1}$ matches the actual next state $\mathbf{s}_{t+1}$.

### UPDATE RULES

Using local synaptic plasticity rules, we update $\mathbf{Q}$ and $\mathbf{V}$ to minimize the prediction error. To do so, we compute the matrices

$$\Delta\mathbf{Q}_{t+1} = \eta_q(\hat{\mathbf{s}}_{t+1} - \mathbf{s}_{t+1})\mathbf{o}_{t+1}^\top \tag{3}$$

$$\Delta\mathbf{V}_{t+1} = \eta_v(\mathbf{s}_{t+1} - \hat{\mathbf{s}}_{t+1})\mathbf{a}_t^\top, \tag{4}$$

where $\eta_q, \eta_v > 0$ are learning rates at each time-step and update $\mathbf{Q} \leftarrow \mathbf{Q} + \Delta\mathbf{Q}$ and $\mathbf{V} \leftarrow \mathbf{V} + \Delta\mathbf{V}$. Iterative updates using these rules can lead to state and action representations consistent with the structure of the underlying environment.

### RANDOM FOURIER FEATURES

Rahimi & Recht (2007) proposed a method to approximate shift-invariant kernels by computing the inner product between vectors produced by a random feature map $\phi$ as a consequence of Bochner's

theorem (Rahimi & Recht, 2007). We describe a mathematically equivalent though slightly different approach in this section commonly used in the Vector Symbolic Architecture (VSA) literature (Plate, 2003; Kleyko et al., 2023).

Suppose we are given vectors $\mathbf{x}, \mathbf{y} \in \mathbb{R}^n$ and a random feature map

$$\phi(\mathbf{x}) = [e^{\sqrt{-1}\mathbf{w}_1^\top \mathbf{x}}, \ldots, e^{\sqrt{-1}\mathbf{w}_D^\top \mathbf{x}}] \in \mathbb{C}^D. \tag{5}$$

where $\mathbf{w}_{ij} \sim p$ for $i = 1, \ldots, D$ and $j = 1, \ldots, n$ for some distribution $p$. We call the output of $\phi$ a random Fourier feature vector. Then the similarity between $\phi(\mathbf{x})$ and $\phi(\mathbf{y})$ is

$$\delta(\phi(\mathbf{x}), \phi(\mathbf{y})) := \frac{1}{D}\text{Re}(\phi(\mathbf{x})^\dagger \phi(\mathbf{y})) \approx K(\mathbf{x} - \mathbf{y}) \tag{6}$$

i.e. it is an unbiased estimate of the evaluation of a shift-invariant kernel $K$ corresponding to the Fourier transform of $p$ at $\mathbf{x} - \mathbf{y}$. $\phi(\mathbf{x})^\dagger$ denotes the Hermitian transpose of $\phi(\mathbf{x})$.

As a special case, if $p$ is the standard Gaussian distribution, then the corresponding kernel $K$ is the Gaussian (or radial basis function) kernel.

Another important property of the random feature map defined in Eq. 5 that we will exploit is

$$\phi(\mathbf{x}) \odot \phi(\mathbf{y}) = \phi(\mathbf{x} + \mathbf{y}) \tag{7}$$

which is a consequence of the additive law of exponents. Here, $\odot$ is the binding operation in VSA, which, in this case, is implemented as element-wise multiplication.

# 3 CMLs in Partially Observable Environments

A CML operates within a fully observable environment; i.e. it assumes that one can exactly infer which state one is in just from a given observation. This assumption is reflected as a bijection between observations $\mathbf{o}_t$ and states $\mathbf{s}_t$: each observation corresponds exactly to one column of $\mathbf{Q}$.

However, most environments do not have this property; they are partially observable. In this section, we introduce an extension of a CML that can operate in partially observable environments, the Partially Observable CML (POCML).

## Two Levels of Representation

The POCML model considers two levels of representation: (1) the "standard" level as in a regular CML, and (2) the "Fourier" level using random Fourier features. This extension of coupled representation is motivated by the need to maintain the straightforward geometric interpretation of CML while enabling the superposition of states.

In our model, we wish to represent uncertainty as a superposition of states. As a naive attempt, we first consider directly superposing the states. If the model is unsure whether it is in state $\mathbf{s}_1$ or $\mathbf{s}_2$, it can represent its estimated state as $\hat{\mathbf{s}}_t = \mathbf{s}_1 + \mathbf{s}_2$. However, if we apply Eq. 2 to predict the next state, we obtain

$$\hat{\mathbf{s}}_{t+1} = \hat{\mathbf{s}}_t + \mathbf{V}\mathbf{a}_t = \mathbf{s}_1 + \mathbf{s}_2 + \mathbf{V}\mathbf{a}_t. \tag{8}$$

This causes an issue both in learning and in interpretation because the superposition of states shares the same operation as a state transition. To address this issue, we propose to use binding instead of addition to predict the next state, i.e.

$$\hat{\mathbf{s}}_{t+1} = \mathbf{s}_t \odot \mathbf{V}\mathbf{a}_t, \tag{9}$$

then we can exploit the distributivity of the binding operation:

$$\hat{\mathbf{s}}_{t+1} = \hat{\mathbf{s}}_t \odot \mathbf{V}\mathbf{a}_t = \mathbf{s}_1 \odot \mathbf{V}\mathbf{a}_t + \mathbf{s}_2 \odot \mathbf{V}\mathbf{a}_t. \tag{10}$$

This approach does not guarantee the same geometric properties intrinsic to the standard representation.

To have the best of both worlds, we apply the random feature map $\phi$ to the standard representation $s$ to obtain a *Fourier* representation $s$. The state superposition and state update of Eq. 10 is applied to

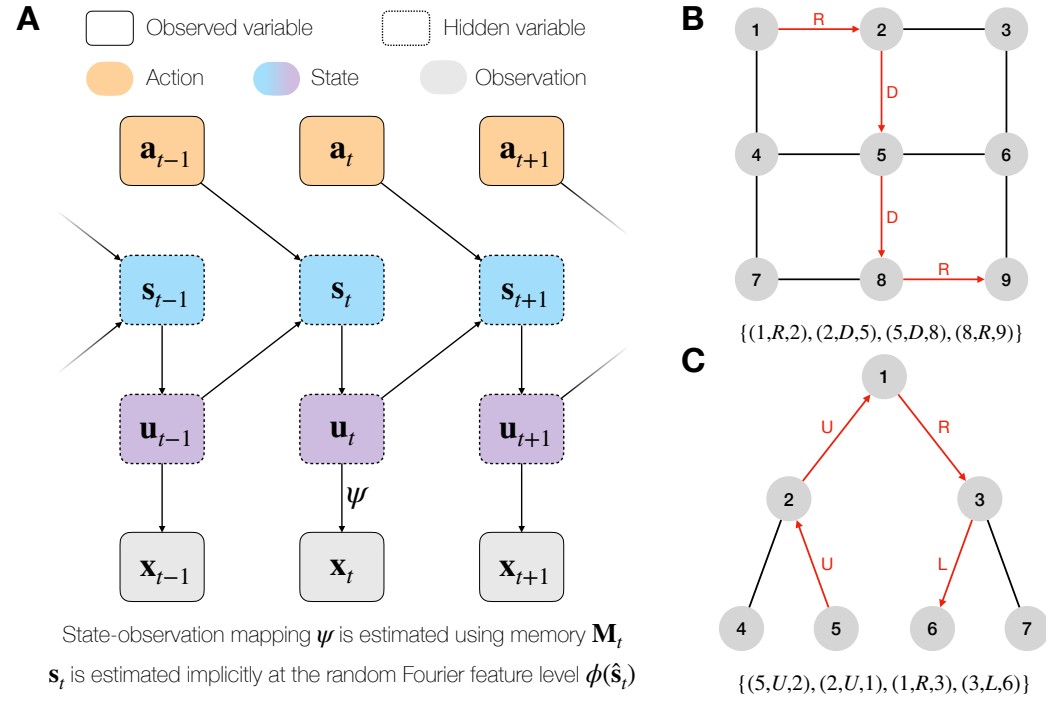

Figure 1: **A**. General structure of the POCML model. **B**. 3-by-3 grid environment with a sample trajectory. **C**. 3-level tree environment with a sample trajectory.

the Fourier representation such that the binding operation has the same interpretation at the standard level thanks to Property 7.

To elaborate, we denote $\hat{\mathbf{s}}_t$ as the estimated superposed state, which can be represented implicitly as a (weighted) set. We extend $\phi$ naturally to the encoding of the set of states via superposition: $\phi(\hat{\mathbf{s}}_t) = \phi(\mathbf{s}_1) + \phi(\mathbf{s}_2) + \delta$ for some noise $\delta$. Then, using Eq.9 to predict the next state, we get

$$\phi(\hat{\mathbf{s}}_{t+1}) = \phi(\hat{\mathbf{s}}_t) \odot \phi(\mathbf{V}\mathbf{a}_t) \tag{11}$$
$$= \phi(\mathbf{s}_1) \odot \phi(\mathbf{V}\mathbf{a}_t) + \phi(\mathbf{s}_2) \odot \phi(\mathbf{V}\mathbf{a}_t) \tag{12}$$
$$= \phi(\mathbf{s}_1 + \mathbf{V}\mathbf{a}_t) + \phi(\mathbf{s}_2 + \mathbf{V}\mathbf{a}_t). \tag{13}$$

This approach is similar to that used in (Kymn et al., 2024). The coupled representation allows us to update all states in superposition while maintaining the desirable geometric properties of the standard CML representation.

## The Model

Suppose, like in a standard CML, an agent explores an underlying environment and produces a sequence of actions and observations.

## Ground Truth and Internal Variables

Actions $\mathbf{a}_t \in \mathbb{R}^{n_a}$, observations $\mathbf{o}_t \in \mathbb{R}^{n_o}$, and states $\mathbf{u}_t \in \mathbb{R}^{n_s}$ are ground truth variables, where $\mathbf{a}_t$ and $\mathbf{o}_t$ are observed (from the environment) while $\mathbf{u}_t$ is unobserved, needing to be inferred. We assume that $\mathbf{o}_t$, $\mathbf{u}_t$, and $\mathbf{a}_t$ are one-hot vectors. We need to distinguish between state and observation here due to breaking the bijectivity (i.e. partial observability) assumption.

Figure 1A visualizes the POCML model structure and the relationship between the different variables. The corresponding standard-level internal model representations are

$$\mathbf{x}_t = f(\mathbf{o}_t) \tag{14}$$

$$\mathbf{u}_t = g(\mathbf{M}_t, \mathbf{x}_t) \tag{15}$$

$$\mathbf{s}_t = \mathbf{Q}\mathbf{u}_t \tag{16}$$

$$\mathbf{v}_t = \mathbf{V}\mathbf{a}_t. \tag{17}$$

where $\mathbf{Q} \in \mathbb{R}^{n \times n_s}$ and $\mathbf{V} \in \mathbb{R}^{n \times n_a}$. $f : \mathbb{R}^{n_o} \to \mathbb{R}^m$ is an arbitrary map. The choice of $f$ depends on the nature of the observation. In this work, as we assume observations to be one-hot, we let $f$ be the identity. Thus, $\mathbf{x}_t$ may be used interchangeably with $\mathbf{o}_t$. The state estimation from the observation is handled by a function over the observation $\mathbf{x}_t$ and a memory unit $\mathbf{M}_t$, discussed in the next section. the same memory is used to estimate the next state (Figure 1A $\psi$).

For the Fourier representation, we choose the random feature map $\phi$ such that it approximates the Gaussian kernel by sampling $\mathbf{w}_{ij} \sim \mathcal{N}(0, 1/\alpha)$; i.e. $\delta(\phi(\mathbf{x}), \phi(\mathbf{y})) \approx \exp(-\alpha\|\mathbf{x} - \mathbf{y}\|^2) = K(\mathbf{x} - \mathbf{y})$. For notational simplicity, let us denote $\mathbf{s}_1, \ldots, \mathbf{s}_{n_s}$ as the columns of $\mathbf{Q}$. Moreover, let $\phi(\mathbf{Q}) = [\phi(\mathbf{s}_1), \ldots, \phi(\mathbf{s}_{n_s})]$.

We perform next-state predictions after applying the random feature map $\phi : \mathbb{R}^n \to \mathbb{C}^D$. As in Eq. 11, we predict the next state via binding:

$$\phi(\hat{\mathbf{s}}'_{t+1}) = \phi(\hat{\mathbf{s}}_t) \odot \phi(\mathbf{v}_t). \tag{18}$$

Here, $\phi(\hat{\mathbf{s}}_t)$ is a linear combination of states $\phi(\mathbf{s}_1), \ldots, \phi(\mathbf{s}_{n_s})$ representing the *expected state* the model is in. Note that $\hat{\mathbf{s}}_t$ is indexed by time while $\mathbf{s}_i$ is indexed by column.

Given the expected state $\phi(\hat{\mathbf{s}}_t)$, we can estimate $\mathbf{u}_t$ via

$$\hat{\mathbf{u}}_t = \mathbb{E}_{p(\mathbf{u}_t|\hat{\mathbf{s}}_t)}[\mathbf{u}_t] = \frac{\mathrm{Re}[\phi(\mathbf{Q})^{\dagger}\phi(\hat{\mathbf{s}}'_t)]}{\sum_{i=1}^{n_s} \mathrm{Re}[\phi(\mathbf{s}_i)^{\dagger}\phi(\hat{\mathbf{s}}'_t)]} \tag{19}$$

Given an observation $\mathbf{x}_t$ and state $\hat{\mathbf{s}}_t$, we can infer the expected state

$$p(\mathbf{u}_t|\hat{\mathbf{s}}_t, \mathbf{x}_t) \propto p(\mathbf{x}_t|\mathbf{u}_t)p(\mathbf{u}_t|\hat{\mathbf{s}}_t) \tag{20}$$

Finally, we let $\phi(\hat{\mathbf{s}}_{t+1})$ be the the superposition of states

$$\phi(\hat{\mathbf{s}}_{t+1}) = \phi(\mathbf{Q})\hat{\mathbf{u}}_{t+1}. \tag{21}$$

HETERO-ASSOCIATIVE MEMORY

We endow the model with a hetero-associative memory

$$\mathbf{M}_t = \sum_{\tau=1}^{t-1} \hat{\mathbf{u}}_t \mathbf{x}_\tau^{\top} \tag{22}$$

where $\mathbf{M}_1 = \mathbf{0}$. Using this memory, given $\hat{\mathbf{u}}_t$, we can predict the observations based on experience:

$$\hat{\mathbf{x}}_t = \mathbb{E}_{p(\mathbf{x}_t|\hat{\mathbf{u}}_t)}[\mathbf{x}_t] = \mathbf{M}_t^{\top}(\hat{\mathbf{u}}_t \odot (\hat{\mathbf{n}}_t^u)^{-1}) \tag{23}$$

where $\hat{\mathbf{n}}_t^u = \sum_{\tau=1}^{t-1} \hat{\mathbf{u}}_\tau$ is a vector recording the expected state counts. In addition, given an observation $\mathbf{x}_t$, we can infer what the state should be based on experience:

$$\tilde{\mathbf{u}}_t = \mathbb{E}_{p(\mathbf{u}_t|\mathbf{x}_t)}[\mathbf{u}_t] = \mathbf{M}_t(\mathbf{x}_t \odot (\mathbf{n}_t^x)^{-1}) \tag{24}$$

where $\mathbf{n}_t^x = \sum_{\tau=1}^{t-1} \mathbf{x}_\tau$ is a vector recording the observation counts.

UPDATE RULES

To perform local prediction, we want to update $\mathbf{Q}$ and $\mathbf{V}$ in order to match $\hat{\mathbf{x}}_{t+1}$ and $\mathbf{x}_{t+1}$. We do this by minimizing their cross entropy given by

$$\mathcal{L} = -\log p(\mathbf{x}_{t+1}|\hat{\mathbf{u}}_{t+1}) = -\log \sum_{i=1}^{n_s} p(\mathbf{x}_{t+1}|\mathbf{u}_i)p(\mathbf{u}_i|\hat{\mathbf{s}}_{t+1}) \tag{25}$$

$$\tag{26}$$

Note that we can rewrite $p(\mathbf{u}_i | \hat{\mathbf{s}}_{t+1})$ as

$$p(\mathbf{u}_i | \hat{\mathbf{s}}_{t+1}) = \frac{\delta(\phi(\mathbf{s}_i), \phi(\hat{\mathbf{s}}'_{t+1}))}{\sum_{i=1}^{n_s} \delta(\phi(\mathbf{s}_i), \phi(\hat{\mathbf{s}}'_{t+1}))} \tag{27}$$

$$\approx \frac{\sum_{j=1}^{n_s} [\hat{\mathbf{u}}_t]_j \exp \psi_{ij}}{\sum_{i=1}^{n_s} \sum_{j=1}^{n_s} [\hat{\mathbf{u}}_t]_j \exp \psi_{ij}}, \tag{28}$$

where we denote $\mathbf{s}'_j = \mathbf{s}_j + \mathbf{v}_t$ and $\psi_{ij} = -\alpha \|\mathbf{s}_i - \mathbf{s}'_j\|^2$. We can make the above approximation as the random feature map $\phi$ approximates the Gaussian kernel. So we can express $\mathcal{L}$ as

$$\mathcal{L} \approx \log \sum_{i=1}^{n_s} \sum_{j=1}^{n_s} [\hat{\mathbf{u}}_t]_j \exp \psi_{ij} - \log \sum_{i=1}^{n_s} \sum_{j=1}^{n_s} p(\mathbf{x}_{t+1} | \mathbf{u}_i)[\hat{\mathbf{u}}_t]_j \exp \psi_{ij} \tag{29}$$

$$\tag{30}$$

Taking the gradient, we get

$$\nabla \mathcal{L} \approx \frac{\sum_{i=1}^{n_s} \sum_{j=1}^{n_s} [\hat{\mathbf{u}}_t]_j \exp \psi_{ij} \nabla \psi_{ij}}{\sum_{i=1}^{n_s} \sum_{j=1}^{n_s} [\hat{\mathbf{u}}_t]_j \exp \psi_{ij}} - \frac{\sum_{i=1}^{n_s} \sum_{j=1}^{n_s} p(\mathbf{x}_{t+1} | \mathbf{u}_i)[\hat{\mathbf{u}}_t]_j \exp \psi_{ij} \nabla \psi_{ij}}{\sum_{i=1}^{n_s} \sum_{j=1}^{n_s} p(\mathbf{x}_{t+1} | \mathbf{u}_i)[\hat{\mathbf{u}}_t]_j \exp \psi_{ij}} \tag{31}$$

$$\approx \sum_{i=1}^{n_s} \sum_{j=1}^{n_s} \left( \frac{\delta(\phi(\mathbf{s}_i), \phi(\mathbf{s}'_j))}{\sum_{i=1}^{n_s} \delta(\phi(\mathbf{s}_i), \phi(\hat{\mathbf{s}}'_{t+1}))} - \frac{p(\mathbf{x}_{t+1} | \mathbf{u}_i) \delta(\phi(\mathbf{s}_i), \phi(\mathbf{s}'_j))}{\sum_{i=1}^{n_s} p(\mathbf{x}_{t+1} | \mathbf{u}_i) \delta(\phi(\mathbf{s}_i), \phi(\hat{\mathbf{s}}'_{t+1}))} \right) [\hat{\mathbf{u}}_t]_j \nabla \psi_{ij}. \tag{32}$$

We do not take the derivative through $p(\mathbf{x}_{t+1} | \mathbf{u}_i)$ as it is computed using a table of expected counts $\mathbf{M}_{t+1}$. Computing the derivative with respect to $\mathbf{Q}$ and $\mathbf{V}$, we get the update rules

$$\Delta \mathbf{Q}_{t+1} = \eta_q \sum_{i=1}^{n_s} \sum_{j=1}^{n_s} \gamma_{ij} [\hat{\mathbf{u}}_t]_j (\mathbf{s}'_j - \mathbf{s}_i) \mathbf{u}_i^\top \tag{33}$$

$$\Delta \mathbf{V}_{t+1} = \eta_v \sum_{i=1}^{n_s} \sum_{j=1}^{n_s} \gamma_{ij} [\hat{\mathbf{u}}_t]_j (\mathbf{s}_i - \mathbf{s}'_j) \mathbf{a}_t^\top \tag{34}$$

where

$$\gamma_{ij} = \frac{p(\mathbf{x}_{t+1} | \mathbf{u}_i) \delta(\phi(\mathbf{s}_i), \phi(\mathbf{s}'_j))}{\sum_{i=1}^{n_s} p(\mathbf{x}_{t+1} | \mathbf{u}_i) \delta(\phi(\mathbf{s}_i), \phi(\hat{\mathbf{s}}'_{t+1}))} - \frac{\delta(\phi(\mathbf{s}_i), \phi(\mathbf{s}'_j))}{\sum_{i=1}^{n_s} \delta(\phi(\mathbf{s}_i), \phi(\hat{\mathbf{s}}'_{t+1}))}. \tag{35}$$

## 4 RESULTS

### NEXT OBSERVATION PREDICTION IN GRID AND TREE ENVIRONMENTS

We test the POCML model on both grid and tree environments. An environment can be represented as a directed graph whose nodes are states and edges are actions. Thus, the underlying structure of an environment is defined by its state-action transitions. Here, we consider environments in which actions in general have the same effect across states (e.g. going up does the same thing in every state of a grid). When traversing the environment, an agent receives different observations in different states defined by a state-observation mapping. Two instances of an environment have the same underlying structure but can have different state-observation mappings. We sample trajectories $\{(\mathbf{x}_t, \mathbf{a_t}, \mathbf{x}_{t+1})\}_{t=1}^{T-1}$ from the environment by choosing an initial state in the environment uniformly randomly, then choosing uniformly random actions. Figure 1A and 1B visualize a 3-by-3 grid environment and a 3-level tree environment respectively along with example trajectories.

Given a collection of trajectories from an environment, we train the POCML model by applying the update rules given in Eqs. 33 and 34. As mentioned in the previous section, doing so minimizes the cross entropy between the predicted and actual observations, effectively performing local next observation prediction.

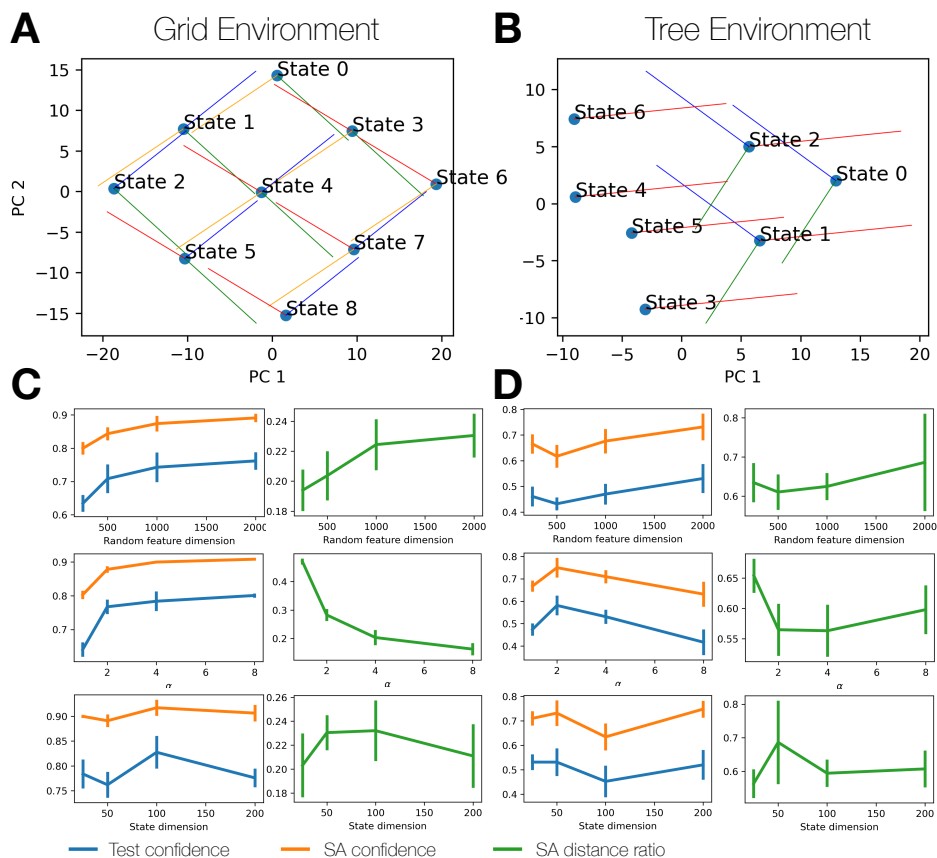

Figure 2: **A**. PCA visualization of learned state and action representations of a POCML trained in a grid environment. **B**. PCA visualization of learned state and action representations of a POCML trained in a tree environment. **C**. Plots of POCML evaluation metrics against various hyperparameters in a grid environment. Error bars report standard deviation over 4 trials. **D**. Plots of POCML evaluation metrics against various hyperparameters in a tree environment. Error bars report standard deviation over 4 trials.

**Learned state representations reflect underlying structure** We perform principal component analysis (PCA) to visualize the learned state representations $\mathbf{Q}$ and action representations $\mathbf{V}$, of the POCML model trained in both grid and tree environments in two dimensions, shown in Figures 2A and 2B respectively. Specifically, the points visualize each state representation projected onto the first two principal components. Actions are visualized as line segments as they represent edges in the underlying graph. They are visualized as the PC projection of the points $(\mathbf{s}, \mathbf{s} + \mathbf{V}\mathbf{a})$ for each state and action in the environment. We find that the arrangement of states in the projected 2D space resembles the underlying structure of the environment. This suggests that the POCML successfully learns the underlying structure of the environment in a self-supervised manner through local next-observation prediction learning.

In addition, we use three evaluation metrics to measure the quality of the learned representations: *next-observation prediction confidence*, *state-action confidence*, and *state-action distance ratio*.

Next-observation prediction confidence measures the average next-observation prediction probabilities over the test trajectories.

State-action confidence measures the quality of state-action transitions by averaging the normalized similarity between the predicted state $\phi(\mathbf{s}) \odot \phi(\mathbf{V}\mathbf{a})$ and the random feature representation of the actual next state $\phi(\mathbf{s}')$ over all state-action pair $(\mathbf{s}, \mathbf{a})$ in the environment.

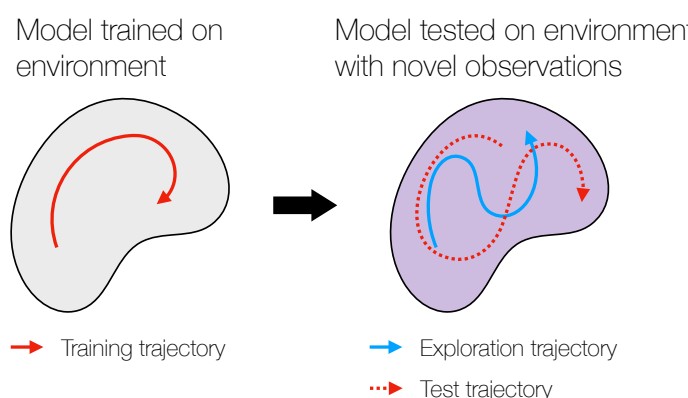

Figure 3: Visualization of zero-shot experimental setup. The model is first trained in an environment by optimizing for next-observation prediction accuracy. The model is tested in a different environment with novel observations. An exploration trajectory is given to populate its memory, while a test trajectory is used to evaluate its next-observation prediction accuracy in the new environment.

State-action distance ratio is similar to state-action accuracy but operates in Euclidean space rather than similarity space (i.e. "standard" level instead of "Fourier" level). The state-action distance ratio is the quantity $\|\mathbf{s}' - (\mathbf{s} + \mathbf{V}\mathbf{a})\|$ normalized by the distance $\|\mathbf{s}' - \mathbf{s}\|$, averaged over all state-action pairs $(\mathbf{s}, \mathbf{a})$ in the environment. A good representation should have high next-observation prediction confidence and state-action confidence as well as a low state-action distance ratio.

Figure 2C and 2D plots the three evaluation metrics for POCMLs with various hyperparameters on grid and tree environments respectively. For each plot, we vary one hyperparameter while keeping the others fixed. The plots show both mean and standard deviation over 4 trials. In the grid environment, we notice that increasing inverse length-scale $\alpha$ and random feature dimension generally leads to better model performance, while there is no clear trend for the state dimension. On the other hand, there is no clear trend in the tree environment. Thus, optimal hyperparameters are environment-dependent.

**POCMLs generalize to environments with different observations in a zero-shot manner**   We test the zero-shot performance of POCML in environments with novel observations. To do this, we generate multiple instances of the environment, where each instance has the same underlying structure (e.g. grid, tree) but with observations that are randomly sampled from a uniform distribution with replacement at each state. For example, in a 3-by-3 grid, we have possible observations $\mathbf{x}_1, \ldots, \mathbf{x}_9$. For each state in the grid, we choose the observation by sampling from $\mathrm{Unif}(\mathbf{x}_1, \ldots, \mathbf{x}_9)$. Thus, the model initially has no information about what observations to expect in these instances. The model must traverse the environment to gain information about the state-observation mapping.

For each instance of an environment, we generate two trajectories. The model uses the first trajectory for exploration in order to populate its memory about the novel environment. The second trajectory is used to test the next observation prediction accuracy of the model. The initial state is provided in both trajectories to ground the model. Note that there is zero training on these new instances so the task is zero shot. Figure 3 describes the zero-shot experimental setup.

We compare the POCML model with both LSTMs (Hochreiter & Schmidhuber, 1997) and Transformers (Vaswani et al., 2017). The LSTMs and Transformers are trained on the same dataset as the POCML for next-observation prediction. Note that we train the LSTM and Transformer models using backpropagation via the Adam optimizer (Kingma & Ba, 2017) while the POCML model only performs local prediction learning via the update rules given above.

Table 1 reports the zero-shot next observation prediction performance of POCML, LSTM, and Transformer on both grid and tree environments. LSTMs and Transformers are chosen to have a comparable number of trainable parameters as the POCML. As shown in the table, the POCML

Table 1: Zero shot performance in environments with novel observations

| Model | Environment | # trainable parameters | Accuracy |
|---|---|---|---|
| POCML | Grid | 450 | **0.980** |
| LSTM | Grid | 493 | 0.121 |
| Transformer | Grid | 451 | 0.117 |
| POCML | Tree | 350 | **0.935** |
| LSTM | Tree | 403 | 0.141 |
| Transformer | Tree | 407 | 0.143 |

model significantly outperforms both LSTM and Transformer models in all respects. This is because the POCML is endowed with strong inductive biases about the environment structure as well as the relation between observation and states, which LSTMs and Transformers lack. Transformers and LSTMs with more (up to $100\times$) parameters yielded similar results.

**POCMLs can disambiguate states with aliased observations**   We consider a two-tunnel maze environment as a case study to investigate model behavior when observations are aliased (i.e. the same observation occurs in two different states). Figure 4A visualizes the two-tunnel maze, while Figure 4B shows how we can model the maze structure and observations using a 3-by-3 grid environment.

We place a goal state in the bottom left corner and provide the agent with a policy $\pi : \mathcal{S} \to \mathcal{A}$ to reach the goal state, where $\mathcal{S}$ is the set of states and $\mathcal{A}$ is the set of actions, shown in Figure 4C. Given an estimated state $\hat{\mathbf{u}}$, the agent acts according to the most likely state it's in $\arg\max_{\mathbf{s}_i \in \mathcal{S}} \hat{\mathbf{u}}_i$. To test whether the POCML model can disambiguate between the two states, we place the agent in the middle-right state and check if it can reach the goal state using the policy. As in the zero-shot experiment above, the agent is given a exploration trajectory to populate its memory.

Figure 4C also shows the trajectory of the agent, while Figure 4D shows a heatmap of the agent's estimated state at each time step $\hat{\mathbf{u}}_t$. Initially, the agent estimated state is a superposition of both left and right tunnel states. After moving down one position, the estimated state collapses to the correct state, and the agent proceeds to take the correct steps to reach the goal state.

## 5   CONCLUSION AND FUTURE DIRECTIONS

In this work, we introduced the Partially Observable Cognitive Map Learner (POCML), an extension of the Cognitive Map Learner (CML) framework to partially observable environments. The POCML distinguishes between two levels of representation and represents states probabilistically through a superposition of states, which are updated via the binding operation. Additionally, we incorporate an associative memory mechanism to support adaptive behavior across environments with shared underlying structures. We derive local update rules for next-observation prediction that account for both the probabilistic state representation and the associative memory.

We demonstrate that a POCML can successfully learn the underlying structure of an environment using these update rules. Furthermore, we show that a POCML trained in one environment generalizes well to new environments with similar structures but novel observations, achieving high zero-shot next-observation prediction accuracy and significantly outperforming models like LSTMs and Transformers. Finally, we perform a case study on spatial navigation in a two-tunnel maze with aliased observations. We show that the POCML effectively leverages its probabilistic state representations for state disambiguation and spatial navigation.

For future work, it is of interest to extend the POCML framework to the continuous domain and adapt the update rules to support a trainable observation encoding component. Given that a POCML effectively learns a cognitive map of the environment, we would also like to investigate its integration with model-based reinforcement learning methods.

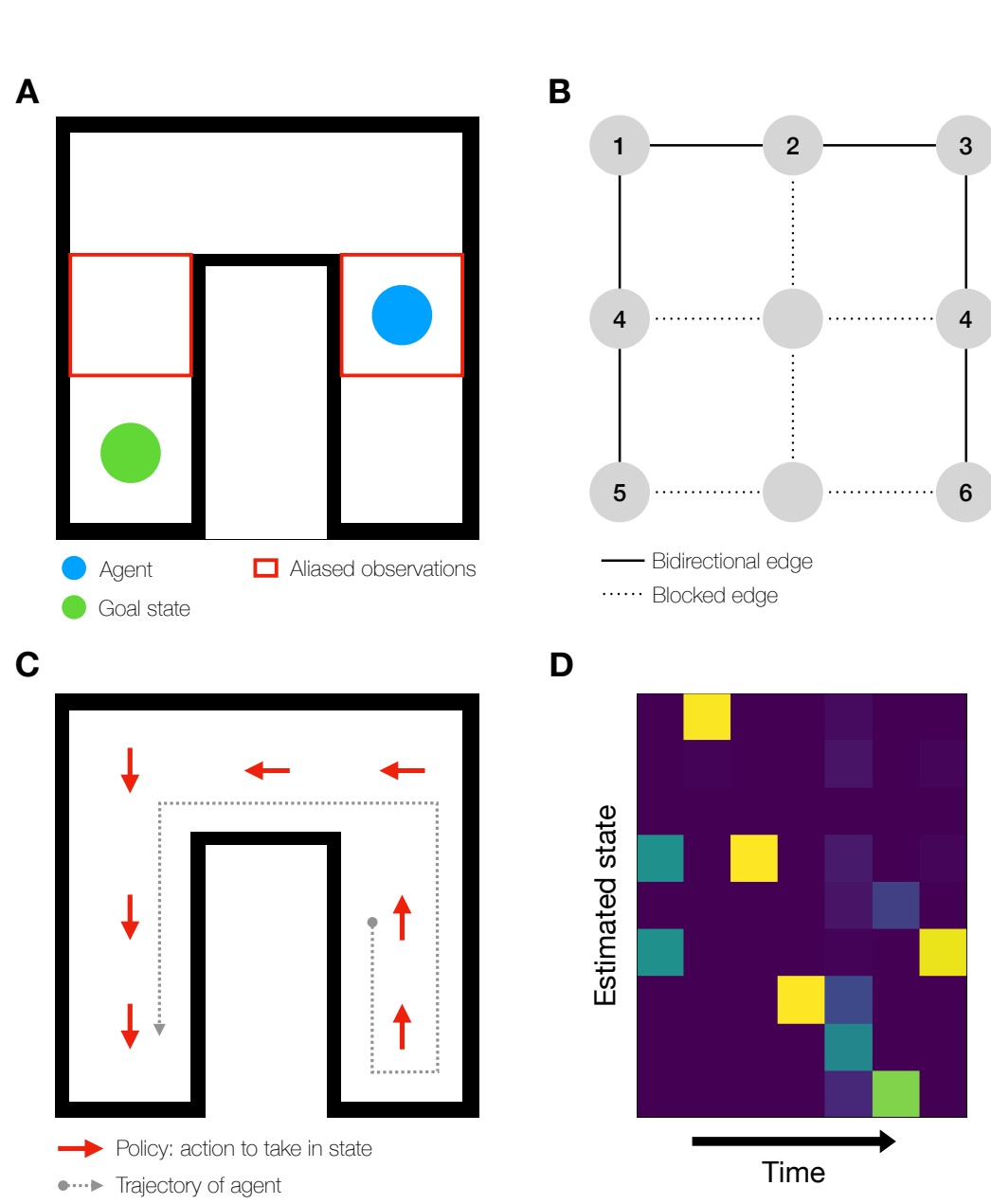

Figure 4: **A**. Two tunnel maze environment. Red boxes indicate aliased observations. **B**. Modeling the two-tunnel maze environment using a 3-by-3 grid environment. Blocked edges simulate walls, while the repeated observation "4" models the aliased observation. **C**. Policy and trajectory visualization of the agent. **D**. Heatmap of the agent's estimated states over the trajectory.

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

## A RELATED WORK

**Comparison against other cognitive map models** TEM is a computational model of the hippocampal-entorhinal system (Whittington et al., 2020). Given sequences of sensory inputs and actions in different environments, it can extract the shared structure between the environments. TEM has two types of latent variables: $g$, which represents the abstract location at a particular point in the sequence, and $p$, which represents a conjunction between sensory observation and abstract location.

TEM and POCML are similar in that both have a separation between abstract state and observation, but TEM uses the conjunctive representation $p$ in its memory component, an auto-associative Hopfield network (Hopfield, 1982). In contrast, POCML uses a hetero-associative memory that associates state and observation implemented as a table of expected counts. In addition, while TEM uses recurrent neural networks (RNNs) to predict the next abstract state, POCML uses the binding operation. While RNNs are more expressive, the binding operation over RFF admits a non-parametric kernel density estimation interpretation of the model. Moreover, the specific choice of Gaussian kernel enables a geometric interpretation of the underlying learned state and action representations: next state prediction can be performed by adding the action representation to the state representation in this space. That said, TEM does learn grid-cell representations that reflect biology and explain the remapping phenomenon; thus, it is valuable from a neuroscience perspective.

From a computational perspective, in both learning and inference, TEM requires iterative processing to perform memory retrieval and state and observation inference, which produces a sizable computational overhead. POCML does not have this limitation.

Algorithmically, there is a comparatively greater difference between CSCG (George et al., 2021) and POCML. Both models can be considered action-augmented hidden Markov models (HMMs) though actions are treated slightly differently in both models. Moreover, CSCG uses a variant of an HMM called a cloned HMM (CHMM). While CSCG represents state-transition probabilities via a state-transition matrix, POCML does this through kernel density estimation (based on RFF). Given that POCML uses vector representations for states and actions, it can easily be extended to larger environments with the same regularity structure simply by keeping the action representations fixed.

**Relation to reinforcement learning** It is important to note that POCML learns the structure of the environment in a reward-free manner, which sets it apart from value-based reinforcement learning (RL) techniques. That said, the structure of the environment learned by the POCML encodes environment dynamics $p(s'|s, a) \propto \delta(\phi(\mathbf{s}'), \phi(\mathbf{s}) \odot \phi(\mathbf{a}))$, which can subsequently be used for model-based RL. The tunnel-maze experiment in the results section demonstrates a way in which this model of the environment can be used given a policy. It is of interest to investigate the consequence of a tighter coupling between model and policy in future work.

## B  LIMITATIONS

This work has several limitations. First, evaluation was performed on simplistic environments. These experiments illustrate the properties and functionality of the POCML model but further work is needed to demonstrate the practicality of the model in larger scale applications. However, we would like to note that the purpose of this work is to introduce the POCML model and the novel representational paradigm based on RFF for encoding uncertainty; we leave extending the model to larger-scale environments for future work. Second, the POCML model here uses discrete observations. Third, as in the original CML work, action affordances have to be provided by the environment.

