# OpenReview forum: "Cognitive map formation under uncertainty via local prediction learning"
_ICLR.cc/2025/Conference — Submitted to ICLR 2025_

### Official Review · Reviewer_VW6F · 2024-10-29

**Soundness:** 3
**Presentation:** 2
**Contribution:** 2
**Rating:** 5
**Confidence:** 4

**Summary:**

This paper presents the "Partially Observable Cognitive Map Learner (POCML)", and extends the Cognitive Map Learner (CML) to handle partially observable environments. To do so, the authors use a Fourier representation that enables to represent state superpositions, and use an associative memory for storing state-observation associations. The resulting model can be optimized using local update rules, and can generalize to environments with the same underlying structure but with novel observations. The authors demonstrate their model on a couple of environments with grid, tree, and two-tunnel maze structure, and compare against LSTM and Transformer models.

**Strengths:**

- The POCML model is sound, and has local update rules which is nice.

- Combining the Fourier features with an associative memory yields good zero-shot performance.

**Weaknesses:**

- The observations are assumed to be one-hot, which is restrictive compared to LSTM/Transformer models which don't have that restriction.

- The introduction mentions other probabilistic model that should enable cognitive map learning in partially observable environments (i.e. TEM and CSCG), and these are set aside as "computationally complex and limited interpretability". However, the authors did not compare any of those on their environments (which I think both TEM and CSCG should also handle well) to harden these claims. Also the "interpretability" of the POCML model is also rather limited (i.e. looking at this PCA visualization).

- The authors should use ~\citep{} to have citations appear as (Behrens et al, 2018).

**Questions:**

- The current experiments are done on rather small environments. How does the method scale to larger environments, e.g. larger grids or larger environments with more structure (e.g. graphs with (hyper)rooms structure as done in George et al)?

-  The authors claim superior interpretability compared to TEM / CSCG? How exactly is it more interpretable? Would a PCA visualization still be interpretable if the environment has more or different structures?

Despite I like the model, I think more and proper experimentation is required, especially comparing against competing approaches in this space (i.e. TEM / CSCG ) as opposed to LSTM/Transformer models which are obviously ill suited for this particular task setup.

---

> ### Author Response · Authors · 2024-12-02
>
> We thank the reviewer for their insightful feedback.
>
> W1. Observations in POCML are assumed to be one-hot as a simplifying assumption for theoretical development and experimental validation. Computationally, the tokenization of state and action can be generalized to account for more general state/action representation. In future work, we are interested in how POCML may model continuous actions and their interpretations.
>
> W2/Q2. While TEM and CSCG are alternative probabilistic models that learn in partially observable environments, their complex computation typically renders the model much more complex than POCML, and the resulting representation of state and action is hard to interpret. In comparison, POCML leverages representation with a simple and unified geometric interpretation of state and action: applying action to a state is equivalent to binding the action vectors to state vectors (at the standard representation). This is further supported by the PCA visualization,  where the geometry learned by the state and action representation matches directly with the underlying environment structure. In comparison, TEM uses recurrent neural networks (RNNs) to predict the next abstract state, and POCML uses the binding operation. Our setup with POCML enables representations in which the underlying geometry is apparent. This holds even in environments of different structures. CSCG is interpretable in the sense that it learns the transition probabilities between different states. However, compared to TEM and POCML, it cannot generalize to larger environments of the same structure as there is no notion of action representations.
>
> W3. Thank you for pointing it out! We have made the changes in the latest draft of the paper.
>
> Q1. Unfortunately, due to time constraints, we were unable to perform experimentation on more standardized environments such as minigrid. However, we would like to point out that we have evaluated Transformer and LSTM models with many more (up to 100x) parameters on the same tasks and they yielded similar results as before. We point this out in the revised paper. We also discussed the issue with the scale of experimentation in a newly added Limitations section in the appendix.

---

> > ### Comment · Reviewer_VW6F · 2024-12-02
> >
> > I thank the authors for their clarifications. After also considering the other reviews and their responses I'm going to stick with my score, as I agree with Reviewer MjwS that the experimental setup is rather on the simple side in the current revision.

---

### Official Review · Reviewer_MjwS · 2024-11-02

**Soundness:** 2
**Presentation:** 2
**Contribution:** 2
**Rating:** 5
**Confidence:** 3

**Summary:**

The paper presents the Partially Observable Cognitive Map Learner (POCML), an extension of the Cognitive Map Learner (CML) for handling partially observable environments. The POCML model incorporates a probabilistic representation of states through superposition, enabling adaptability across environments with similar structures. This is achieved by leveraging a hetero-associative memory and local update rules to support state estimation and environment-specific learning. The authors demonstrate the model’s performance in a highly simplified 3x3 grid and tree environment, with results showing POCML’s superiority over LSTM and Transformer models for next-observation prediction in these toy cases. Additionally, POCML is tested in a two-tunnel maze, showcasing its capability in handling state ambiguity.

**Strengths:**

One of the primary strengths of this paper is the novel extension of CML for partially observable settings, introducing an interesting formulation of the POCML model. By integrating probabilistic state representations and associative memory, POCML provides a creative approach to modeling cognitive maps under uncertainty. This formulation demonstrates the potential for enabling adaptive behavior across structurally similar environments, which is an intriguing step toward more robust and flexible spatial representation models.

**Weaknesses:**

The paper suffers from several critical limitations that undermine its contributions. The primary issue is its reliance on an overly simplistic experimental setup—a 3x3 grid and a basic tree structure. This setting, being extremely limited in scale, does not convincingly support the model’s general applicability or superiority. For a method proposed to handle partially observable cognitive map learning, demonstrations in mildly complex environments are essential. In this toy setup, where built-in inductive biases favor POCML, it is unsurprising that the model would outperform general sequence models like LSTMs and Transformers. This limitation severely restricts the external validity of the findings.

Moreover, the experimental comparisons with LSTMs and Transformers are performed on models with fewer than 500 parameters, a scale that is unlikely to allow for meaningful generalization in neural model, especially when paired with such a naive problem. The small scale of these comparison models likely leads them to overfit, rendering the observed performance differences unconvincing. The experimental results, therefore, do not robustly establish the claimed superiority of POCML over these general architectures.

Besides, I doubt the effectiveness of limited size assumption on state space, action space, and observation space. In most interesting applications, space size should be much larger.

**Questions:**

See above.

---

> ### Author Response · Authors · 2024-12-02
>
> Thank you for the insightful feedback.
>
> **Simplistic experimental setup**: We acknowledge the need for more complex environments. While the experimental environments presented in this work are relatively simple in comparison with the common setup in TEM and CSGS, they are purposefully chosen to evaluate the foundational capabilities of POCML as an extension of CML. These setups enable rigorous testing of the model's ability to learn structure and disambiguate observations in partially observable settings, a critical challenge for cognitive map formation. In particular, we followed the practice in CML [1] to test the POCML in the grid and tree structures, which are representative abstractions of more complex environments. Because we assume a partially observable setting, the problem complexity increases significantly from that with the same underlying topology in a CML experiment, leading to reduced-size grid and tree environments for our experimentation. For future work, we aim to validate POCML in larger-scale and more diverse settings, such as 3D mazes or complex graph environments.
>
> **Transformers/LSTM fewer than 500 parameters**: We agree with the reviewer that the main results (Table 1) of our comparison with Transformers and LSTM use a small number of parameters, though we would like to point out that we have evaluated Transformer and LSTM models with many more (up to 100x) parameters on the same tasks and they yielded similar results as before. The parameter choice in Table 1 is intended to investigate the model performances with a similar number of trainable parameters. We have emphasized this in the revised paper.
>
> [1] Stöckl, Christoph, Yukun Yang, and Wolfgang Maass. "Local prediction-learning in high-dimensional spaces enables neural networks to plan." *Nature Communications* 15.1 (2024): 2344.

---

> > ### Comment · Reviewer_MjwS · 2024-12-02
> >
> > I have read the authors' response. However, I do not personally find it convincing. I'm still under the impression that the tested environments are to simple to warrant fair comparison with potentially more power models, and setups.

---

### Official Review · Reviewer_orX1 · 2024-11-03

**Soundness:** 3
**Presentation:** 2
**Contribution:** 2
**Rating:** 6
**Confidence:** 3

**Summary:**

This paper describes an extension of a cognitive map learning model (CML) published in Nature Communications earlier this year to partially observed graphs. CML is a neural network model that learns to predict next states from actions, such that the trained system constitutes a cognitive map. States and actions are described by vectors in \mathbb{R}^n, e.g. in a 2-D space each state/observation is specified by (x, y) coordinates.
This cognitive map is showed to support resource-rational online planning in contexts of graphs where reasonably good plans can be achieved through a heuristic of taking actions in the direction of the goal. Here the direction toward of the goal is given by computing similarity between vectors describing pairs of states. For example, Euclidean distance provides the direction assuming states are spatial locations. Obviously this model is imperfect, as the "closest" state may be not directly reachable, and may require a circuitous indirect route, while biological brains efficiently handle such situations. These model are proof of concept neural implementations of a cognitive map to support online planning, however they are neither behaviorally validated nor supported by theoretical justifications from neuro or behavioural literature, such as discussion of vector navigation in biological organisms.

The current paper extends the CML system to represent states in a probabilistic manner by using random Fourier features. This model is called a partially observable CML (POCML). The model is demonstrated to work in a simple toy example. It is a somewhat hard for me to see the value of the contribution, as I am looking for either a scientific research question that we learn about, or a potential for engineering application. At the same time, I acknowledge the value of building a novel system using an original methodology as contributing to the engineering spirit of the conference, and recommend acceptance for this reason.

**Strengths:**

The work presented in the paper is novel, and the approach is original.

**Weaknesses:**

These points maybe partly an issue of space available for a conference submission, but this paper comes across as somewhat lacking motivation and clarity.

Mainly the motivation of the paper was unclear to me. "Cognitive maps are central to the adaptive behavior of intelligent agents..." suggests biological agents by linking to a Tim Behrens paper in Neuron. Likewise "While we do not know how exactly the brain implements cognitive maps..." suggests a biological model, or a biologically-inspired system. However the authors do not attempt to use the model to replicate behavior, neither by discussing empirical behavioral principles that could be predicted by this model, nor by fitting it to experimental data. Perhaps the motivation is to built a biologically-inspired model for neuromorphic hardware, or to improve on existing model-based reinforcement learning, but it is not presented that way.

I had to read the original CML paper to understand what is going on, I wonder if Supplementary Information could be useful to compensate for the lack of space.

Limitations are not discussed.

**Questions:**

The example on which the model is demonstrated is very much toy. Is there something like a practical application domain for this model? If not, what is the value of this toy example, what is the scientific research question it is demonstrating?

---

> ### Author Response · Authors · 2024-12-02
>
> Thank you for the positive review and the insightful feedback.
>
> **Unclear Motivation**: Thank you for pointing out the ambiguity in our motivation. The main goal of our paper is to extend the existing Cognitive Map Learner (CML) algorithm to handle partially observable environments, a setting adopted by concurrent theoretical models of cognitive map formation, resulting in the proposed Partially Observable Cognitive Map Learner (POCML).
>
> **Lack of Limitation**: Thank you for pointing out the lack of limitation sections in our work. We have followed your advice and added a limitations section to the appendix. This work has several limitations. First, evaluation was performed on simplistic environments. These experiments illustrate the properties and functionality of the POCML model but further work is needed to demonstrate the practicality of the model in larger scale applications. However, we would like to note that the purpose of this work is to introduce the POCML model and the novel representational paradigm based on RFF for encoding uncertainty; we leave extending the model to larger-scale environments for future work. Second, the POCML model here uses discrete observations. Third, as in the original CML work, action affordances have to be provided by the environment.
>
> **Experimental setup**: We acknowledge the need for more complex environments. While the experimental environments presented in this work are relatively simple in comparison with the common setup in TEM and CSGS, they are purposefully chosen to evaluate the foundational capabilities of POCML as an extension of CML. These setups enable rigorous testing of the model's ability to learn structure and disambiguate observations in partially observable settings, a critical challenge for cognitive map formation. In particular, we followed the practice in CML [1] to test the POCML in the grid and tree structures, which are representative abstractions of more complex environments. Because we assume a partially observable setting, the problem complexity increases significantly from that with the same underlying topology in a CML experiment, leading to reduced-size grid and tree environments for our experimentation. For future work, we aim to validate POCML in larger-scale and more diverse settings, such as 3D mazes or complex graph environments.

---

### Official Review · Reviewer_M3BT · 2024-11-04

**Soundness:** 3
**Presentation:** 4
**Contribution:** 3
**Rating:** 5
**Confidence:** 4

**Summary:**

The paper extends an existing cognitive architecture called Cognitive Map Learning (CML) that was recently proposed as an explanation for cognitive map formation and planning in the biological brain via prediction errors. The original CML architecture is shown to be applicable to only fully observable environments. This paper aims to directly address the drawback while maintaining efficiency and geometric interpretability. Their method, called PO-CML brings together two mathematical tools to address the issues with CML: (1) superposition of states using random Fourier features, for a probabilistic state interpretation (2) Associative Memory to enable transfer of behavior across environments with similar structures. Experiments are performed on toy grid and tree environments with 9 and 7 states respectively. The learned state and action representations can be seen via 2D PCA projections, having structures identical to the original environments. This shows that cognitive maps are learnt in partially observable environment via prediction errors alone. Experiments are done to test zero-shot transfer to new environments against Transformers and LSTMs with 500 parameters. POCML quickly learns the cognitive map for the newer environments. PO-CML is also tested on a 3x3 grid with aliased observations and is shown to distinguish between them based on context despite observing same inputs. This shows that PO-CML successfully extends the CML architecture to partially observable environments.

**Strengths:**

1. Originality
    - This paper builds on existing CML [1] idea, and introduces new operations that can enable CML to function effectively in Partially Observable environments, which is an important and relevant extension. Prior work is well established, and the claimed scope and novelty is clear.
    - The new introduction of binding operation and superposition of states is innovative and well justified for the problem of mapping similar observations to several possible candidate states. I think there is some takeaway even for the RL folks in this formulation, since typically in RL and robotics an RNN is assumed to absorb the partially observable inputs.
2. Quality
    1. Clever connections between CML formulation and Random Fourier Features. Superposition of states is well justified due to the non-bijective nature of Partial observability.
    2. The simple experiments are relevant and further help understand the necessity of PO-CML. But as I mention below, I find the experiments insufficient to make serious conclusions.
    3. I want to highlight that I found the writing to be very crisp, precise and to the point. I really enjoyed reading the paper.
3. Clarity
    1. Well explained idea, with rigorous formulation and precise language.
    2. All the inputs, outputs and assumptions are made explicit in the experiments, making it easy to understand.
4. Significance
    1. Very relevant topic in Cognitive Architectures, Computational Neuroscience, Reinforcement Learning and Planning research.
    2. In future, CML and their extended architectures can potentially explain planning activity in the biological brain and also hint AI researchers to use more efficient architectures.

**Weaknesses:**

1. **Limited Experimentation**: My biggest concern with the paper is the overly simplistic experiments performed to establish the superiority of PO-CML. The environments considered are a grid with 9 states and a tree with 7 states. This is small even for a cognitive architecture paper. While I think the simplicity of the experiments make the idea clear and neat, I am also a realist and believe that the community will take PO-CML seriously, when it can be demonstrated on more complex scenarios. The predecessor paper [1] uses a larger graph and several other domains like mujoco-ant to establish the algorithm. Would it be possible to use such familiar and benchmarked environments for the paper in addition to existing experiments? Minigrid [4] is one possible suggestion.
2. **Experiment setup**: Adding on to (1), I want a discussion on the following questions: Why were the particular environments chosen for evaluation? Does PO-CML scale like transformers when used on more complex environments with even more data? What would be other relevant scenarios where directly applying PO-CML would benefit me? What are some other design considerations when choosing the PO-CML architecture for other POMDP problems like [4]?
3. **Ablation** against transformers and LSTMs with more parameters: Table 1 reports comparison agains LSTM and Transformers with ~ 500 parameters which is probably the smallest transformer I have seen. I also think that this comparison is not fair/relevant. Architectures like Transformers shine when the parameter count and dataset increase to very large numbers. Would it be possible to test the accuracy of transformers/LSTMs/PO-CML as you scale the number of parameters and possibly data?
4. **Explanation of Mathematical Tools used**: The authors assume strong familiarity of all the mathematical tools used in the paper. I was somewhat lost in page 3 when they introduce vector symbolic architectures and mention the property $\phi(x) \odot \phi(y) = \phi(x + y)$. Please cite more relevant works and mention why that property holds true. A derivation in the appendix section for the above property would be helpful too.
5. **Comparison against existing work**: While the authors qualitatively compare CML against TEM and CSCG in the introduction section, they do not compare their own work against it. I would request an in-depth discussion of algorithmic and technical novelty when compared to the other recent works.
6. **Choice of Baselines**: To make my point (3) more clear, why not use CML, TEM and/or CSCG as baselines, since the authors compare CML against them and they clearly have trade-offs that are relevant for the computational neuroscience and cognitive architecture community.
7. **Qualitative comparison** against value-based RL, model based RL, hierarchical RL approaches: Several works in model based RL and value based methods tackle the problem of learning value functions in a given environment. A value function is very similar to a cognitive map with the possibility of online planning once the environment is explored. Hierarchical RL also extends value functions to successor-representations that can be transferred across environments [5][6]. Can the authors discuss and compare CML against these approaches?

**Questions:**

Below, I explicitly highlight the clarifications and details I need to make a final call regarding the scores. The below concerns distill the points mentioned in weakness section, and are the reasons why I lean towards a borderline score at this point. Authors, please directly address the issues below. I like the paper and **I am willing to increase my score towards an accept/strong accept if these concerns are addressed appropriately during the discussion phase**. I have gone through the relevant figures and explanations, but please point out the exact page/table/figure, if I have missed a detail that’s already provided in the paper.

1. **Request for Experiments**: Note that I understand the time constraints during the rebuttal phase and have made best efforts in making sure my requests are not overblown/broad/irrelevant to the central claim of the paper.
    - Address weakness (1): Learn a model on one other POMDP environment, preferably something like [4] which is established, has several baselines and is highly reproducible. I leave the exact choice of environment to the authors. Simple instantiations of the environment and non-image, token only inputs are ok at this point. Results for the new environment can be plotted as PCA visualization similar to Figure 2 in the paper. The goal here is for another researcher in comp neuro or RL to quickly understand what the state and actions spaces are and be able to adopt and experiment with PO-CML.
    - Address weakness (3): Ablations for zero-shot next observation prediction on the new environment. Please ablate against increasing number of parameters for all the models listed in Table 1.
    - Address weakness (6): Original CML as a baseline to highlight the issues with non-bijective partially observable states on the newly chosen environment above.
    - Address weakness (6): TEM or CSCG as a baseline on the new environment, to highlight the tradeoffs/improvements.
2. **Request for Discussion:**
    - Address weakness (2), (4), (5), (7): Please discuss the respective questions mentioned in the weakness section directly. This can be added to the paper/appendix in appropriate sections if the authors find it to be informative to the readers.
3. **PCA Figure clarification**: Can the authors clarify what the exact inputs and outputs to the model in Figure 2 are? Why is it partially observable? How was the figure generated? Why are the states represented as points in space and actions represented by “line segments” (instead of points/lines)?

### **Other concerns and comments**

1. What exactly is the binding operation introduced in page 3 to avoid issues with addition? Since it is not a trivial operator, can the authors clarify this in the paper?
2. Page 2 Equation $(3)$ and $(4)$, is the $\Delta$ meant for gradient operator? Please replace it with the more commonly used $\nabla$ symbol. Otherwise, please clarify.
3. Correction: Figure 1B and 1C on Page 6 should be Figure 1A and 1B.
4.  No explicit reproducibility statement: I’d encourage the authors to commit to reproducibility of their work with code/environment release since I do not see that in the paper at the moment.

### References

[1] Christoph Stockl, Yukun Yang, and Wolfgang Maass. Local prediction-learning in high dimensional spaces enables neural networks to plan. *Nature Communications*

[2] James C. R. Whittington, Timothy H. Muller, Shirley Mark, Guifen Chen, Caswell Barry, Neil Burgess, and Timothy E. J. Behrens. The Tolman-Eichenbaum Machine: Unifying Space and Relational Memory through Generalization in the Hippocampal Formation.

[3] Dileep George, Rajeev V. Rikhye, Nishad Gothoskar, J. Swaroop Guntupalli, Antoine Dedieu, and Miguel La ́zaro-Gredilla. Clone-structured graph representations enable flexible learning and vicarious evaluation of cognitive maps

[4] https://github.com/Farama-Foundation/Minigrid

[5] [https://elifesciences.org/articles/78904#:~:text=The SR postulates a predictive,frequently being represented more strongly](https://elifesciences.org/articles/78904#:~:text=The%20SR%20postulates%20a%20predictive,frequently%20being%20represented%20more%20strongly).

[6] https://awjuliani.medium.com/the-present-in-terms-of-the-future-successor-representations-in-reinforcement-learning-316b78c5fa3

---

> ### Author Response · Authors · 2024-12-02
>
> We sincerely thank the reviewer for the detailed constructive feedback. It has been immensely helpful to us. We have tried to follow your feedback in the revised copy of the paper and for what we could not immediately apply we will definitely consider for our future work.
>
> Unfortunately, due to time constraints, we could not perform experimentation on more standardized environments such as minigrid. However, we would like to point out that we have evaluated Transformer and LSTM models with many more (up to 100x) parameters on the same tasks and yielded similar results.
>
> Regarding weakness (2), POCML works best in environments with regular structures (e.g. grids and trees), as action and state representations are decoupled. This is because, taking after TEM, the motivating example used in developing the model is to learn spatial structure. So we are mostly concerned with environments of that type here, explaining our choice of environments, which are also used in the TEM paper. To improve model performance in more complex environments, we plan to extend it in future work.
>
> Based on your feedback, to address weakness (5), we added a related work section in the appendix comparing in detail POCML against TEM and CSCG.
>
> Specifically, TEM and POCML are similar in that both have a separation between abstract state and observation, but TEM uses the conjunctive representation $p$ in its memory, a Hopfield network. In contrast, POCML uses a hetero-associative memory that associates state and observation implemented as a table of expected counts. TEM uses recurrent neural networks (RNNs) to predict the next abstract state while POCML uses the binding operation. While RNNs are more expressive, properties of binding enable a geometric interpretation of the learned representations: next state prediction can be performed by adding the action representation to the state representation in this space. That said, TEM does learn grid-cell representations that reflect biology and explain the remapping phenomenon; thus, it is valuable from a neuroscience perspective.
>
> From a computational perspective, in both learning and inference, TEM requires iterative processing to perform memory retrieval and state and observation inference, which produces a sizable computational overhead. POCML does not have this limitation.
>
> Algorithmically, there is a comparatively greater difference between CSCG and POCML. Both models can be considered action-augmented hidden Markov models (HMMs) though actions are treated slightly differently in both models. Moreover, CSCG uses a variant of an HMM called a cloned HMM (CHMM). While CSCG represents state-transition probabilities via a state-transition matrix, POCML does this through kernel density estimation (based on RFF). Given that POCML uses vector representations for states and actions, it can easily be extended to larger environments with the same regularity structure simply by keeping the action representations fixed.
>
> We also discuss the relation to RL here to address weakness (7), which we also include in the appendix. It is important to note that POCML learns the structure of the environment in a reward-free manner and the resulting state and action representation is independent of policy, which sets it apart from value-based reinforcement learning (RL) techniques. That said, the structure of the environment learned by the POCML encodes environment dynamics $p(s'|s,a)\propto \delta(\phi(\mathbf{s}'),\phi(\mathbf{s})\odot\phi(\mathbf{a}))$, which can subsequently be used for model-based RL. The tunnel-maze experiment in the results section demonstrates a way in which this model of the environment can be used given a policy. It is of interest to investigate the consequence of a tighter coupling between model and policy in future work. As you point out, Successor Representations (SR) have been used in RL as a reward-free representation of states. However, SRs consider distributions over future states $s’$ at any given state $s$ via $M(s,s’)$, given a policy so they are still policy-dependent. It has been pointed out that SRs need to be recomputed for dynamic planning [A]. It is also unclear how it can generalize beyond known states as it lacks action representations. POCML representations, conversely, can be used to generate a policy. One could potentially connect SRs and POCML representations by considering the relation between $M(s,s’)$ and $\sum_a p(s'|s,a)p(a)$ (from above), but this still needs further exploration.
>
> To address weakness (4), we have added an explanation of the binding operation and why this property holds in the background section.
>
> We have also clarified both PCA figure in the results section and that $\Delta$ does not refer to an operator; $\Delta Q$ simply refers to the matrix that we add to $Q$ when applying the update rule.
>
> [A] Stachenfeld, K. L., Botvinick, M. M. & Gershman, S. J. The hippocampus as a predictive map. Nat. Neurosci. 20, 1643–1653 (2017).

---

### Meta-Review · Area_Chair_xoLp · 2024-12-20

**Metareview:**

This paper introduces PO-CML, an extension of the Cognitive Map Learning (CML) architecture that enables it to learn cognitive maps in partially observable environments. PO-CML uses random Fourier features and associative memory to address CML's limitations. Experiments on toy problems (grid and tree environments with 5-10 states) demonstrate successful map learning and zero-shot transfer to new environments, and is compared to transformers and LSTMs (although with only 500 parameters). With these experiments, the paper demonstrates that PO-CML successfully extends CML to the partially observable setting.

Reviewers particularly praised the clarity of the paper's presentation -- the writing was clear, and the contribution was well scoped. Everyone also agrees that the idea is original, using Fourier features to tackle the partially observable setting for CML is generally agreed to be a clever idea.

However, reviewers also pointed out limitations with the experiments. The environments investigated here are extremely simple -- 5-10 states is a much smaller environment than any AI / ML agent would normally face. Similarly, the comparison to transformers and LSTMs does not really make sense when only 500 parameters are allowed in the architectures -- these models are generally used with much larger parameter counts and access to significant data. While I agree that there were time restrictions and so significant extra experiments would not be possible, I do not think it would have been unreasonable to test the PO-CML model on the same grid / tree- structured environments with larger state spaces. This should have been a simple extension that would be easy to test during the rebuttal period, but these experiments were not run (despite being requested by reviewers).

For a future submission, running the PO-CML model on larger environments would help demonstrate the effectiveness of the method. Without these experiments, I do not think the paper will be impactful enough for an AI / ML audience, and therefore recommend rejection.

**Additional Comments On Reviewer Discussion:**

Reviewers requested extra experiments expanding the environments to include more states / actions, which were not provided during the rebuttal period.

Reviewers also requested better contextualization of the method with respect to prior works such as TEM, which the authors did incorporate into a revised version of the paper (albeit in the appendix). However, several reviewers also commented on the need for explicit comparison to TEM (for example, how interpretable the representations are), which the authors allude to but do not include in the revision.

For a future submission, adding TEM as a baseline to the paper and extending to more complex environments seems key to make this paper compelling for an AI / ML audience.

---

### Decision · Program_Chairs · 2025-01-22

Reject